# A Heterogeneity-Aware Semi-Decentralized Model for a Lightweight Intrusion Detection System for IoT Networks Based on Federated Learning and BiLSTM

**DOI:** 10.3390/s25041039

**Published:** 2025-02-09

**Authors:** Shuroog Alsaleh, Mohamed El Bachir Menai, Saad Al-Ahmadi

**Affiliations:** 1Department of Computer Science, King Saud University, Riyadh 11451, Saudi Arabia; 2Department of Information Technology, Princess Nourah bint Abdulrahman University, Riyadh 11671, Saudi Arabia; menai@ksu.edu.sa (M.E.B.M.); salahmadi@ksu.edu.sa (S.A.-A.)

**Keywords:** internet of things, intrusion detection system, anomaly detection, machine learning, deep learning, federated learning, energy-based, centralized FL, decentralized FL, semi-decentralized FL, transfer learning

## Abstract

Internet of Things (IoT) networks’ wide range and heterogeneity make them prone to cyberattacks. Most IoT devices have limited resource capabilities (e.g., memory capacity, processing power, and energy consumption) to function as conventional intrusion detection systems (IDSs). Researchers have applied many approaches to lightweight IDSs, including energy-based IDSs, machine learning/deep learning (ML/DL)-based IDSs, and federated learning (FL)-based IDSs. FL has become a promising solution for IDSs in IoT networks because it reduces the overhead in the learning process by engaging IoT devices during the training process. Three FL architectures are used to tackle the IDSs in IoT networks, including centralized (client–server), decentralized (device-to-device), and semi-decentralized. However, none of them has solved the heterogeneity of IoT devices while considering lightweight-ness and performance at the same time. Therefore, we propose a semi-decentralized FL-based model for a lightweight IDS to fit the IoT device capabilities. The proposed model is based on clustering the IoT devices—FL clients—and assigning a cluster head to each cluster that acts on behalf of FL clients. Consequently, the number of IoT devices that communicate with the server is reduced, helping to reduce the communication overhead. Moreover, clustering helps in improving the aggregation process as each cluster sends the average model’s weights to the server for aggregation in one FL round. The distributed denial-of-service (DDoS) attack is the main concern in our IDS model, since it easily occurs in IoT devices with limited resource capabilities. The proposed model is configured with three deep learning techniques—LSTM, BiLSTM, and WGAN—using the CICIoT2023 dataset. The experimental results show that the BiLSTM achieves better performance and is suitable for resource-constrained IoT devices based on model size. We test the pre-trained semi-decentralized FL-based model on three datasets—BoT-IoT, WUSTL-IIoT-2021, and Edge-IIoTset—and the results show that our model has the highest performance in most classes, particularly for DDoS attacks.

## 1. Introduction

With the number of IoT devices significantly increasing, the need to preserve the security and privacy of transmitted data has become a challenging task. Many mechanisms have been used to secure IoT networks, for example, firewalls and access-control mechanisms, but these methods only ensure the confidentiality and authenticity of data within the network of IoT devices. Thus, IoT networks need security mechanisms that can serve as a line of defense for detecting intruders. An intrusion detection system (IDS) needs to be applied to detect malicious behaviors. Two common approaches for intrusion detection systems (IDSs) are signature-based intrusion detection systems (SIDSs) and anomaly-based intrusion detection systems (AIDSs). An SIDS, that is, “heuristics-based or misuse detection”, defines a set of known malicious data patterns (signatures) or attack rules (heuristics) that are used for attack detection. However, this approach can only identify known attacks for which it has stored patterns or rules for comparison. On the other hand, an AIDS, that is, “behavior-based IDS”, entails developing the profiles of normal users’ behaviors over a period of time so that any deviation from this profile will be viewed as an anomaly or abnormal behavior [1].

Most IoT devices have limited resource capabilities to carry out conventional IDS tasks. Therefore, lightweight IDS approaches have been proposed to fit IoT device capabilities. First, energy-based IDSs are based on detecting attacks by using energy consumption as an indicator of specific types of attacks. However, not all attack types can be detected using this method because many attacks have no impact on energy consumption (e.g., Sybil or data manipulation attacks) [2,3]. Second, machine learning and deep learning (ML/DL) models are used to analyze data and their dependencies to help recognize attack patterns [4]. Moreover, ML/DL-based IDSs are used as lightweight IDSs by extracting the most relevant features for attack detection. However, with the number of interconnected IoT devices increasing dramatically and generating a massive amount of data, ML/DL models are difficult to deploy within resource-constrained IoT networks [5,6].

One ML model, known as federated learning (FL), is a distributed ML model that enables IoT devices to learn a shared ML model collaboratively without disclosing local data. Thus, the introduction of FL in IDSs helps decrease the computational burden of central processing servers while preserving data privacy. Three FL architectures are commonly used: centralized FL (client–server); decentralized FL (device-to-device); and semi-decentralized FL. In the centralized FL architecture, each device trains the local model on its own data, and then, its parameters/weights are aggregated to produce a global model controlled by a central entity (i.e., a server or coordinator) [7,8]. Even though centralized FL improves the efficiency of the learning process, it has some drawbacks, namely a lack of scalability and connectivity and a single-point-of-failure vulnerability. Decentralized FL overcomes these issues by allowing devices to collaborate in the learning process without a coordinator. However, the aggregation process is inefficient because it is not conducted under the supervision of one device, resulting in low accuracy [9,10,11]. Semi-decentralized FL strikes a balance by clustering devices, with each cluster having a local coordinator to facilitate learning and aggregation, resulting in faster convergence and improved scalability [12,13,14].

In all FL architectures, heterogeneity is a core challenge that hinders FL performance [15] because heterogeneity greatly impacts the learning process, resulting in model divergence [16,17,18]. The resource-constrained nature of IoT devices demands lightweight models. Furthermore, most current approaches fail to achieve a balance between high performance and efficient resource utilization.

In this paper, we present a semi-decentralized FL model to detect intrusions in a heterogeneous IoT network. The primary contributions of our work are outlined below:The proposed semi-decentralized FL model clusters FL clients to eliminate the impact of heterogeneity in the FL learning process and reduce resource consumption.The clustering approach in the proposed model helps in improving the performance of FedAvg, the aggregation algorithm, as each cluster sends the average model’s weights to the server for aggregation in one FL round.We investigated three DL techniques, LSTM, BiLSTM, and WGAN, as local models in the semi-decentralized FL architecture. To the best of our knowledge, we are the first to apply WGAN with one generator for all FL clients to train them on the same generated data since our target is to detect intrusion, not to discriminate the real data from the generated data. Each FL client has a discriminator model to train it on its local data.The semi-decentralized FL model with BiLSTM strikes a balance between performance and lightweight-ness to fit the IoT capabilities as it is configured with low-complexity parameters (number of layers and neurons).

The remainder of this paper is organized as follows: Section 2 reviews related work. Section 3 provides a detailed explanation of the proposed solution. Section 4 presents the experimental evaluation and discusses the results along with a comparative study. Finally, Section 5 concludes the study by summarizing the findings and highlighting open research directions.

## 2. Related Work

As IoT devices have limited resource capabilities, there is a need to consider this limitation when designing IDSs for the IoT. Many approaches have been proposed to develop lightweight IDSs, namely energy-based IDSs, ML/DL-based IDSs, and federated learning-based IDSs.

### 2.1. Lightweight Energy-Based IDSs

Since conventional IDSs are not applicable for resource-constrained IoT devices, many studies have focused on minimizing energy consumption through lightweight IDSs. Krontiris et al. [2] and Riecker et al. [3] proposed an energy-based approach that emphasizes analyzing energy consumption to identify attacks by estimating expected energy consumption based on past observations for each device. Thus, if a given device’s energy use deviates from the expected value, it is viewed as an anomaly. Unfortunately, this approach cannot detect all types of attacks, only those that impact energy consumption, for example, various types of DoS attacks (flooding, blackholes, etc.).

### 2.2. Lightweight Machine Learning/Deep Learning-Based IDSs

Machine learning and deep learning (ML/DL) techniques are commonly used in IDSs to analyze data and their dependencies as a way to learn attack patterns [2]. In ML, feature engineering is performed to extract the relevant features for attack identification and classification. Many ML algorithms are used in IDS, for example, decision tree (DS), support vector machine (SVM), logistic regression (LR), naïve Bayes (NB), and artificial neural network (ANN). However, DL algorithms can automatically extract high-level features through multiple non-linear processing layers. According to the studies by Ahmad et al. [5] and Thakkar et al. [6], the most commonly used DL models in IDSs include convolutional neural network (CNN), recurrent neural network (RNN), long–short-term memory (LSTM), and autoencoder (AE). To create lightweight IDSs for the IoT environment, ML/DL-based IDSs are used for feature selection and dimensionality reduction [19,20,21]. However, because of the significant growth in interconnected IoT devices, which generate huge amounts of data, ML/DL-based IDSs have become difficult to deploy in resource-constrained IoT networks.

### 2.3. Lightweight Federated Learning-Based IDSs

FL is a privacy-preserving distributed ML model that enables FL clients to learn a shared ML model collaboratively without disclosing local data. The application of FL for IoT networks has been used recently, as the ML model is distributed among FL clients, which helps in preserving the computation resources. Three FL architectures are commonly used: centralized FL (client–server), decentralized FL (device-to-device), and semi-decentralized FL.

#### 2.3.1. Centralized FL (Client–Server)

In this architecture, each device independently trains its local models using its own data. The parameters or weights from these local models are then aggregated to form a global model, which is managed by a central entity, such as a server or coordinator, as shown in Figure 1. The training process involves several rounds. In each round, clients send their weights to the server, which updates the global model. This process continues until the desired accuracy is achieved or a predetermined number of rounds is completed [7,8].

The deployment of centralized FL for IoT networks comes in two schemes: first, the edge–cloud architecture, in which a cloud server functions as an FL server and an edge device functions as an FL client; second, the edge layer architecture, in which the edge device functions as the FL server and the IoT device (end node) functions as the FL client. Many studies have been proposed based on cloud–edge architecture. Huong et al. [22] developed a low-complexity cyberattack detection in IoT edge computing (LocKedge), in which the cloud server extracts the most relevant features using the principal component analysis (PCA) method, and the NN model’s hyper-parameters and initial weights can be set. This information is then sent to all edge devices to train their models with their own data using SGD. Once all edge devices finish the training process, they send the updated weights of their model to the server for the aggregation process. LocKedge is evaluated in two modes, traditional ML and FL, and the results showed that traditional ML has a higher detection rate than FL because of uneven data distribution—non-IID (non-independent and identically distributed among edge devices). Rashid et al. [23] developed an FL-based model for cyberattack detection in IIoT networks. They applied CNN and RNN for both traditional ML and FL. According to their results, the FL model with a low number of rounds and the traditional ML model differ considerably (both CNN and RNN). However, as the number of FL rounds increases, the margin shrinks, and at the 50th round, the FL-based model reaches the same intrusion detection accuracy as traditional ML. Similar work is presented in [24] but with datasets distributed in a different ratio among clients. Their results show that the variation in dataset distributions would be the cause of the performance gap between FL and traditional ML. This is because the non-IID data could cause local updates from different clients to clash, which results in performance degradation of the global model. Zhang et al. [25] proposed a federated learning framework known as FedDetect for IoT cybersecurity. The proposed model has improved the performance by utilizing a local adaptive optimizer and a cross-round learning rate scheduler instead of FedAvg for local training. According to the evaluation results of two settings—traditional ML and FL—FL’s detection accuracy is worse than traditional ML because, with the former, the server cannot directly learn the data’s features, as with traditional ML, making FL worse for feature learning. However, the system efficiency analysis indicates that both end-to-end training time and memory costs are affordable and promising for resource-constrained IoT devices. Another FL-based IDS model was developed by [26] using a deep autoencoder to detect botnet attacks using on-device decentralized traffic data. They installed a virtual machine on each edge device to conduct the local model training, and they used port mirroring on the edge devices so that network traffic flow toward the IoT devices was not interrupted. However, using a virtual machine comes with many issues, such as complexity, less efficiency, and high cost. An ensemble, multiview, FL-based model was proposed by [27], which categorizes packet information into three groups: bidirectional features, unidirectional features, and packet features. Each view is trained using NN, and then their predicted results are sent to the rain forest (RF) classifier, which acts as an ensembler that combines the three views’ predictions and provides a single attack prediction based on its probability and occurrence. Similarly, Driss et al. [28] also proposed FL-based attack detection that uses RF to ensemble the global ML models—GRU with different window sizes—to detect attacks in vehicular sensor networks (VSN). According to the experimental results of both studies [27,28], the use of an ensembler unit increases the attack prediction accuracy. Friha et al. [29] applied FL in an agricultural IoT network using three DL models—DNN, CNN, and RNN— on three different datasets. Their results show that one of the datasets—namely InSDN—outperformed the traditional ML model. The results also demonstrate that time and energy consumption have been affected by the DL model and number of clients involved in the FL process.

The communication burden is one of the biggest issues in centralized FL as each client sends their model weights/parameters to the server and then receives the new weights/parameters (after aggregation) from the server. Thus, a client selection is proposed to reduce the number of participating clients in the learning process. An FL-based model deployed in the edge layer, in which the edge device is the FL server and the IoT device is the FL client, was proposed by [30,31]. Considering that IoT devices vary in their resource capabilities, one that satisfies the requirements of ML/DL algorithms must be selected. Chen et al. [30] developed an asynchronous FL model to improve training efficiency for heterogeneous IoT devices. The selection algorithm is based on the node’s computing resources and its communication condition. Thus, instead of waiting for all nodes to send their weight vectors to be aggregated by the server, only the selected clients can send their updates, which accelerates the learning process in its convergence. Rjoub et al. [31] developed a client selection method based on two metrics—resource availability (in terms of energy level) and IoT devices’ trustworthiness—to make appropriate scheduling decisions. They measure the IoT device’s trust level (trust score) based on utilizing the resources (over/under a specific threshold value) during the local training. Even though the client selection mechanism helps in improving the learning process, it may affect the prediction accuracy, where not all clients are involved in the learning process. On the other hand, Chen et al. [32] proposed communication-efficient federated learning (CEFL), which assures communication efficiency and resource optimization that differ from [30,31]. Specifically, a client selection method is proposed based on the calculation of the gradient difference between the current model parameter and the model parameter from the previous round. Then, the client sends its new local model only if the gradient difference is large; otherwise, the server will reuse the stale copy of that client. Even though the CEFL algorithm has succeeded in reducing the communication burden, it adds additional computational overhead for calculating the gradient difference.

Transfer learning (TL) is combined with FL to improve learning performance. Nguyen et al. [33] introduced a new compression approach—known as high-compression federated learning (HCFL)—to handle FL processes in a massive IoT network. HCFL applies TL to enhance learning performance by training a pre-model with a small amount of the dataset on the server. Abosata et al. [34] also incorporated TL with FL to develop accurate IDS models with minimum learning time. The main goal of the TL model is to transfer the local parameters generated using gradient-based weights from the clients to the server. Their experimental results show that their model succeeded in minimizing the complexity of the learning model and maximizing the accuracy of global model generation. Nguyen et al. [35] were the first researchers to employ FL in anomaly-based IDSs; they developed a device-type-specific model in which IoT devices are mapped to the corresponding device type for anomaly detection because IoT devices that belong to the same manufacturer have similar hardware and software configurations, resulting in highly identical communication behavior. Wang et al. [36] applied the same device-type-specific model to develop an IDS model for each type of IoT device, which improves the model’s accuracy. It also addresses IoT device heterogeneity in terms of communication protocols and coexisting technologies. However, this approach cannot be applied in large-scale IoT networks where there are various types of IoTs that need to be mapped, which will result in delay. Tabassum et al. [37] developed the first FL-based IDS utilizing GAN to train local data, addressing class imbalance by generating synthetic data to balance the distribution across FL clients. They evaluated their model using different datasets, with the results showing that the proposed model converged earlier than the state-of-the-art standalone IDS. Bukhari et al. [38] proposed an FL-based IDS using a stacked CNN and bidirectional long–short-term memory (SCNN-Bi-LSTM). They addressed the statistical heterogeneity of the data by applying a personalized method that could divide the model into global and local layers. Specifically, the global layers are used in the FL, and each FL client is provided with the global layers with unique customization layers designed for their specific needs. The experimental results show a significant improvement in detection accuracy compared with traditional DL approaches.

#### 2.3.2. Decentralized FL (Device-to-Device)

The decentralized FL architecture relies on device-to-device (D2D) communication, where devices collaboratively train their local models using stochastic gradient descent (SGD) and consensus-based methods. During each consensus step, devices share their local model updates with their one-hop neighbors. Each device then integrates the model updates received from its neighbors and inputs the results into the SGD process [9,10]. In 6G networks, devices are anticipated to be interconnected in a machine-to-machine (M2M) manner, making D2D communication a promising technology for decentralized FL [7,8]. Figure 2 shows the decentralized FL architecture.

Although the centralized FL has demonstrated success in the IoT environment in terms of preserving computation resources as well as improving the learning process, it includes some drawbacks, that is, single-point-of-failure and scaling issues for increasing network size. This leads to the FL being fully distributed (server-less), in which the end devices collaborate to perform data operations inside the network by iterating local computations and mutual interactions. Savazzi et al. [9] proposed a decentralized FL-based consensus approach for IoT networks that can enable direct device-to-device (D2D) collaboration without relying on a central coordinator. This approach has been viewed as a promising framework for IoT networks because of its flexible model optimization over networks characterized by decentralized connectivity patterns. However, they applied their methods to a simple NN, which would be difficult to apply to deeper networks. A trusted decentralized FL-based algorithm was integrated with the consensus approach in [10]. Their results indicate that the trusted decentralized FL algorithm makes the model robust against poisoning attacks. Blockchain federated learning (B-FL) based on a consensus approach was proposed in [11] to prevent model tampering from malicious attacks in both local and global models. The B-FL system comprises multiple edge servers and devices that generate a blockchain based on a consensus protocol to confirm that the data in the block are correct and immutable. They applied digital signatures to guarantee that others were not tampering with the data packet information. Although digital signatures ensure the authenticity and integrity of the data, they require computation overheads that are beyond the IoT devices’ capabilities.

Al-Abiad et al. [39] proposed a decentralized FL model based on device-to-device (D2D) communications and overlapped clustering to allow aggregation in a decentralized scheme without a central aggregation. The proposed model is based on the overlap between clusters. Specifically, within a cluster, the cluster head (CH) is associated with each cluster. On the other hand, the bridge device (BD) is located between two adjacent clusters. The BDs receive the updated models from the CHs and calculate the summation of the received aggregated models. As a result, CHs collect all the local models from the corresponding devices and models from the BDs to update their cluster models. Their experimental results show that the proposed model outperformed the centralized and hierarchical FL models in terms of energy consumption and accuracy, here considering efficient overlapped clustering and increasing the number of iterations, respectively.

#### 2.3.3. Semi-Decentralized FL

The semi-decentralized FL architecture aims to strike a balance between fully centralized and fully decentralized FL architectures. This approach employs multiple servers, each responsible for coordinating a group of client devices to perform local model updates and inner model aggregation. Periodically, each server shares its updated models with neighboring servers for outer model aggregation. This multi-server aggregation process, as opposed to relying on a single centralized entity, helps to speed up the learning process [12,13,14]. Figure 3 illustrates the semi-decentralized FL architecture.

A semi-decentralized approach was developed by [12] to accelerate the learning process by considering multiple edge servers. Each edge server coordinates a cluster of client nodes to perform local model updating and intra-cluster model aggregation. The edge servers periodically share their updated models with neighboring edge servers for inter-cluster model aggregation. Even though their model accelerates the learning process, it poses computation overhead on the edge server because it needs to perform aggregation for the cluster and for the neighboring servers. To overcome this problem, a hierarchical architecture was presented in [13,14] with two levels of model aggregation: the edge and cloud levels. More specifically, the edge node is responsible for local aggregation of clients, whereas the cloud facilitates global aggregation because it has a reliable connection. Both studies’ experimental results demonstrate a reduction in the model training time and energy consumption of the end devices when compared to traditional cloud-based FL. The author of [40] introduced hierarchical FL based on semi-synchronous communications to solve the heterogeneity of IoT devices. The proposed model performs edge-based aggregations and then cloud-based aggregation, resulting in a significant communication overhead reduction. This is because edge-based aggregation is performed via local communications, which require fewer resources than cloud-based aggregation. Moreover, a semi-synchronous communication in both the cloud and edge layers is proposed to prevent the deadlock that comes as a result of waiting for local updates from dropped devices. Thus, a timeout is defined such that if all local models are not received by this timeout, then the edge server sends the latest aggregated edge model to the cloud server without waiting for the remaining local models to arrive. Huang et al. [41] proposed a semi-decentralized cloud–edge-device hierarchical federated learning framework, incorporating an incremental sub-gradient optimization algorithm within each ring cluster to mitigate the effects of data heterogeneity. This hierarchical FL architecture not only reduces communication overhead but also enhances overall performance.

Alotaibi and Barnawi [42] developed IDSoft, an innovative software-based solution that operates throughout the network infrastructure, using 6G technologies such as virtualization of network functions, mobile edge computing and software-defined networking. This solution is designed to support FL-based IDSs. Additionally, they designed a hierarchical federated learning (HFL) model within IDSoft to detect cyberattacks in 6G networks. Synchronous and asynchronous schemes are used in the aggregation process. The results show that asynchronous HFL has faster convergence and stable training loss compared with the centralized FL model. In addition, the HFL model achieves lower communication overhead than the centralized FL model because the centralized FL architecture requires communication between each client and the server for aggregation; however, in HFL, clustering helps in reducing the total communication load by allowing the CH to act on the behalf of the clients. However, they did not mention other hierarchical FL-based IDSs, which may indicate that they were the first to introduce this model. The same authors developed LightFIDS (Lightweight and Hierarchical Federated IDS for Massive IoT in 6G Networks) [43], a module within the IDSoft architecture they previously introduced. In IDSoft, there are two detection phases: a fast detection phase that identifies abnormal flows at the far edge, followed by a deep detection phase that provides further analysis of malicious flows. The proposed LightFID model is implemented with fast-converging HFL. The local learning models used in the federated architecture are low-complexity models with fewer parameters. However, the hierarchical federated learning architecture is more complex, requiring the cluster head to perform aggregation for multiple rounds in order to obtain the cluster model. The head master (server) also aggregates the received cluster models to get the global model. This process is computationally intensive due to the repeated aggregation and distribution within clusters.

### 2.4. Motivation

From all the above, FL with the three architectures has succeeded in designing IDS for IoT networks, but with some drawbacks. In centralized FL, communication latency is the main issue due to the time required to collect the trained model from clients, especially considering the heterogeneity of IoT devices. Another significant issue is communication overhead, as all clients must communicate directly with a single server, leading to increased network load and energy consumption [10,30]. The decentralized FL relies on the D2D transmission, which has a low network cost and serves as a practical solution for faster convergence. This, in turn, reduces delay and energy consumption. Additionally, decentralized FL resolves the single-point-of-failure and scalability issues inherent in centralized FL. However, it has a major drawback: the aggregation process is inefficient, just as it is in the centralized FL model, because aggregation is not performed under the supervision of one device, resulting in an accuracy drop-off [9,10,11].

Semi-decentralized FL solves the issues of centralized and decentralized FL architectures. Specifically, the semi-decentralized FL model is based on clustering the FL clients and then assigning a cluster head (CH) for each cluster that communicates with the FL server. Therefore, clustering will help in reducing the total communication load by allowing the CH to act on behalf of the FL clients, which helps in preserving the resource consumption (bandwidth) [42]. Moreover, the clustering can also be used to solve data heterogeneity, which is considered one of the biggest issues of deploying FL in IoT networks [14,25]. Clustering clients based on received model updates after training the FL clients was proposed by [44,45,46]. According to their theoretical findings, the cosine similarity between different clients’ weight updates is highly indicative of their data distributions’ similarities. After clustering the clients, each cluster trains its model independently until the global model for each cluster converges. However, most proposed semi-decentralized FL approaches primarily focus on addressing data heterogeneity and minimizing communication overhead, without considering the lightweight structure of the model, specifically its size (i.e., the number of parameters).

## 3. Proposed Solution

This section presents the proposed IDS model for IoT networks. It begins with the description of how FL clients are generated, considering all types of attacks. Following that, it presents how the semi-decentralized FL is applied after clustering the FL clients. Next, it provides an overview of the deep learning techniques, namely LSTM, BiLSTM, and WGAN, that serve as local models within the FL architecture.

### 3.1. Generating FL Clients

First of all, the dataset will be preprocessed and split into training, validation, and testing sets, as will be explained in the experimental evaluation in Section 4. The training set is then distributed across the FL clients, considering that each client should have all types of attack as well as benign data. Since the dataset’s classes are imbalanced, we need to consider this during the dataset distribution by taking a ratio from each class based on the class size (number of instances for each type of class). The distribution of the dataset is carried out randomly; thus, we generate five different clients’ distributions, known as “runs”. Algorithm 1 presents the method to generate FL clients.
**Algorithm 1:** CreateClient.
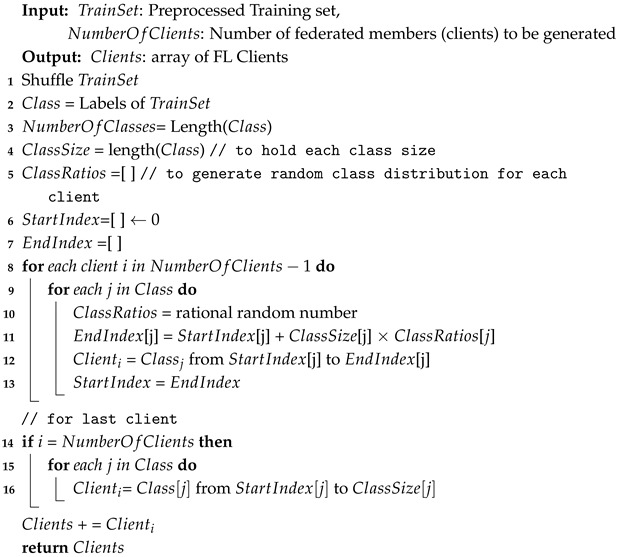


### 3.2. Proposed Semi-Decentralized FL Model

The proposed semi-decentralized federated learning (FL) model clusters clients based on the model updates received from FL clients. Specifically, each client is trained using a simple Multi-Layer Perceptron (MLP) for one round as a preprocessing step before clustering. Next, the updated model weights are collected from each client to prepare for clustering. Since the model weights are high-dimensional, dimensionality reduction is applied before clustering the clients to avoid sparsity issues in the clusters. Alkhayrat et al. [47] conducted a comparative study between principal component analysis (PCA) and autoencoder (AE) for dimensionality reduction before clustering. Their results indicate that AE outperforms PCA in their case study. Therefore, AE is applied to the received model weights to reduce their dimensions. The reduced weights are then scaled before applying the clustering algorithm [48]. To identify the most suitable clustering algorithm, we investigate various algorithms to determine the best one for our study. Yin et al. [49] reviewed and analyzed existing clustering algorithms based on key indicators such as time complexity, scalability, and clustering quality. According to their comparative study, K-means is the most suitable clustering algorithm for our case, as it efficiently handles large datasets with low time complexity. To apply k-means clustering, the number of clusters needs to be defined as well as the distance metrics. According to the experimental results, the k-means with Manhattan distance achieves the highest Silhouette score results, as will be explained in Section 4.3.3. A cluster head (CH) is chosen for each cluster based on the averaged Silhouette score. The Silhouette score is used because it indicates whether a client is correctly positioned within its assigned cluster and not too close to the boundary of neighboring clusters. Algorithm 2 summarizes the clustering methodology that we developed.
**Algorithm 2:** Clustering.
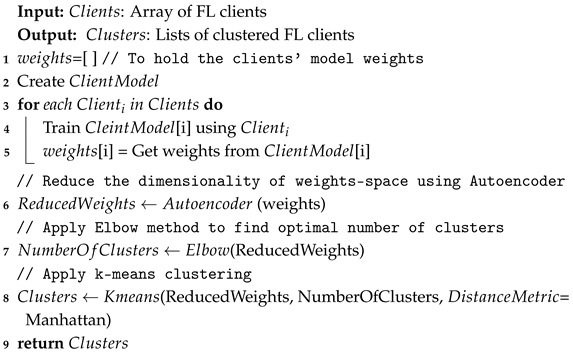


The semi-centralized FL model starts after clustering the clients and assigning the CH for each cluster. The server broadcasts the model with initial weights and parameters to the CH, then the CH sends them to all clients within its cluster. Each client then trains its local model and sends the updated model to its corresponding CH. Once the CH receives the updated model from all clients, it calculates the average weights and sends them to the server. After the server obtains all new weights from each cluster, it aggregates the received model weights from all CH and calculates the global model using FedAvg, as shown in Equation (Equation 1). Then, the server broadcasts the new model to the cluster heads to update their local model. This process is repeated for a given number of rounds or until the model converges, where no significant improvement in performance is observed. To evaluate performance during training, the global model generated after each round is validated using the validation set, and performance metrics are calculated. Algorithms 3–5 summarize the semi-decentralized model, demonstrating the training procedure on the server side, cluster side, and client side, respectively. After completing the FL training, the global model is tested on the testing set to evaluate its performance. Algorithm 6 presents the FL testing process. Figure 4 illustrates the proposed semi-decentralized FL-based architecture.
**Algorithm 3:** Semi-decentralized FL server training.
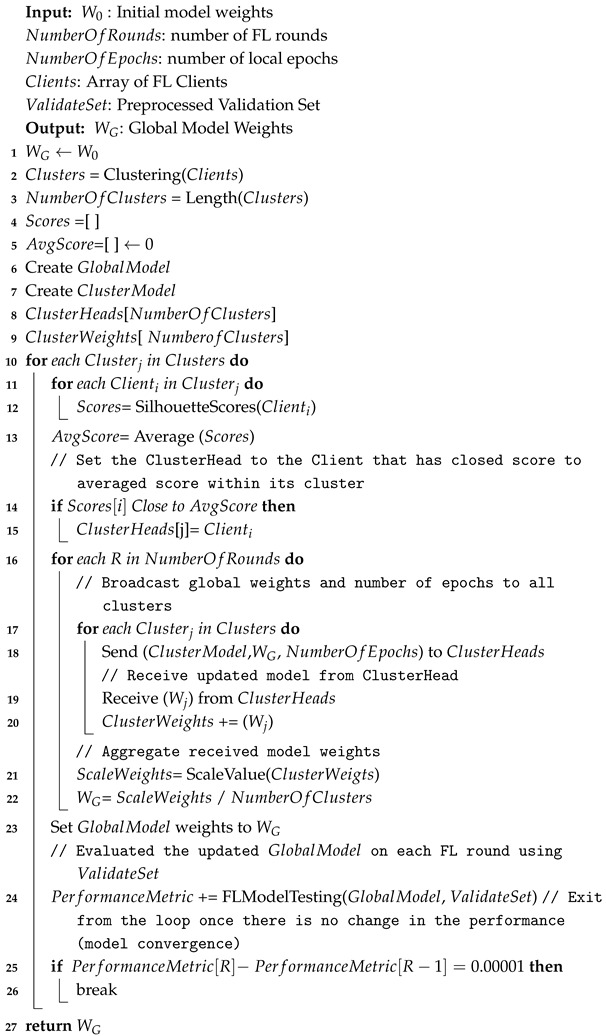


**Algorithm 4:** Semi-decentralized FL cluster training.
  **Input**: $W_{G}$: Global Model Weights,   NumberOfEpochs: Number of Local Epochs  **Output**: $W_{j}$: Cluster Model Weights**_1_** Receive ClusterModel from the server **_2_** Receive initial weights and parameters from the server **_3_** Send ClusterModel, weights, and parameters to the clients **_4_** $W_{i}$ += Get model weights from clients  //
( Average the received client weights within the same cluster
**_5_** 

(


$W_{j}\leftarrow \mathrm{Average}\left(W_{i}\right)$

**_6_** **return** $W_{j}$


**Algorithm 5:** Semi-decentralized FL client training.
  **Input**: $W_{G}$: Global Model Weights,   NumberOfEpochs: Number of Local Epochs  **Output**: $W_{i}$: Client Model Weights**_1_** Receive ClientModel from the ClusterHead
**_2_** Receive weights and parameters from the ClusterHead ($W_{G}$, NumberOfEpochs) **_3_** Train each ClientModel with received weights and parameters **_4_** $W_{i}\leftarrow$ Get weights from ClientModel  //
( Return updated client weights $W_{i}$ to the ClusterHead for aggregation (**_5_** **return** $W_{i}$


**Algorithm 6:** FL model testing.
  **Input**: GlobalModel: Global Model after FL process   TestSet: Preprocessed testing set  **Output**: DetectionRate: metric that used to measure model performance**_1_** XTest= Features of TestSet**_2_** YTest= Labels of TestSet**_3_** PredictedY = Test GlobalModel on XTest
**_4_** Calculate PerformanceMetric based on PredictedY and YTest**_5_** **return** PerformanceMetric


### 3.3. Aggregation Algorithm

Once all clients have trained their data, each cluster head collects the new parameters/weights from its clients, calculates the average weights, and sends them to the server for aggregation. The server then calculates the aggregated weights by averaging the received weights/parameters using the FedAvg algorithm [50] as shown in the following formula:(1)$$w_{\mathrm{global}}=\sum_{k=1}^{K}\frac{n_{k}}{n}w_{k}$$

$w_{\mathrm{global}}$ is the global model’s weights.*K* is the number of clusters.$n_{k}$ is the total number of data samples in each cluster *k*.*n* is the total number of data samples in all clusters, $n=\sum_{k=1}^{K}n_{k}$.$w_{k}$ are the weights of the model trained in the cluster *k*.

Equation (Equation 1) shows that the global model weights are a weighted sum of the cluster’s model weights, with the weights proportional to the number of data samples in each cluster. The FedAvg is used at the server level, considering that each cluster sends the average weights of its clients to the server, unlike other semi-decentralized approaches [43] where the aggregation process is performed in two levels: the cluster head and the server. The cluster head conducts the aggregation process for multiple rounds to obtain a cluster model. Then, the server aggregates the received cluster models to obtain the global model and sends it back to the cluster heads for the next FL round. This process is computationally intensive due to the repeated aggregation and distribution within clusters. Our proposed model reduces this complexity by allowing each cluster head to collect weights from its clients, average them once, and send the averaged weights directly to the server. The server then aggregates the received cluster’s models and returns the global model to the cluster heads, all within one FL round, thereby reducing the communication overhead and making it affordable for resource-constrained IoT.

### 3.4. Local Deep Learning Models

#### 3.4.1. Long–Short-Term Memory (LSTM)

LSTM is an extension of recurrent neural networks (RNNs) with the ability to solve the problem of vanishing gradients in an efficient way. LSTM enhances RNNs’ memory capabilities, allowing them to learn long-term dependencies from inputs. This extended memory can store information over longer periods, enabling the model to read, write, and delete information as needed. The LSTM memory is structured as a “gated” cell, meaning it can decide whether to retain or discard information. This gating mechanism allows LSTM to capture essential features from inputs and maintain this information over extended sequences. The decision to keep or remove information depends on the weights assigned during training, allowing the model to learn which details are worth preserving or discarding. An LSTM model typically consists of three gates: the forget gate, input gate, and output gate, as shown in Figure 5. The “forget gate” decides which information should be retained or discarded. It uses a sigmoid function to evaluate both the current input $x_{t}$ and the previous hidden state $h_{t-1}$. The output of the forget gate, represented by $f_{t}$, ranges between zero and one. A value close to zero indicates that the information will be discarded, while a value near one implies that it will be retained. This mechanism is formulated as(2)$$f_{t}=\sigma \left(W_{f}\cdot h_{t-1},x_{t}+b_{f}\right)$$
where $W_{f}$ represents the weight and $b_{f}$ the bias for the forget gate, and σ denotes the sigmoid activation function.

The “input gate” then determines which new information should be added to the cell state. It also uses a sigmoid function, which result in values from zero (unimportant) to one (important). The input gate’s function is defined as(3)$$i_{t}=\sigma \left(W_{i}\cdot h_{t-1},x_{t}+b_{i}\right)$$

Next, the tanh function processes the current input $x_{n}$ and the previous hidden state $h_{t-1}$, generating a vector of new candidate values, $C_{t}$, which may be added to the cell state:(4)$$\tilde{C_{t}}=\tanh\left(W_{c}\cdot h_{t-1},x_{t}+b_{c}\right)$$
The cell state $C_{t}$ is updated by first multiplying the previous cell state $C_{t-1}$ by $f_{n}$, and then adding the product of $i_{t}$ and $C_{t}$ where ⊙ denotes element-wise multiplication, and tanh is the hyperbolic tangent activation function.(5)$$C_{t}=\left(f_{t}⊙C_{t-1}+i_{t}⊙\tilde{C_{t}}\right)$$

Lastly, the “output gate” determines the next hidden state $h_{t}$. It takes into account both the cell state and the previous layer’s output:(6)$$o_{t}=\sigma \left(W_{o}\cdot h_{t-1},x_{t}+b_{o}\right)$$(7)$$h_{t}=o_{t}⊙\tanh\left(C_{t}\right)$$

In these equations, $W_{o}$ and $b_{o}$ are the weights and bias associated with the output gate. The output gate employs the sigmoid function σ and the tanh function to compute the final output $h_{t}$, which represents the next hidden state [38,51].

#### 3.4.2. Bidirectional Long–Short-Term Memory (BiLSTM)

BiLSTM is an extension of the LSTM model where two LSTM models are applied to the input data. Unlike the standard LSTM network, which only utilizes the information it has already encountered in the sequence, the BiLSTM architecture incorporates two LSTM layers—one processing the input sequence forwards (forward LSTM) and the other processing it backward (backward LSTM), as shown in Figure 6. This dual application of the LSTM significantly enhances the model’s capabilities to learn long-term dependencies, ultimately leading to improved performance [38].

#### 3.4.3. Wasserstein Generative Adversarial Network (WGAN)

Wasserstein GAN (WGAN) is an alternative to the traditional GAN model. In GAN, there are two neural networks—the generator and the discriminator—that engage in a dynamic adversarial process. The generator aims to capture the data distribution and then produce synthetic data that look like the real data. On the other hand, the discriminator evaluates incoming samples and predicts whether they originate from the real dataset or were generated synthetically. This competition drives both networks to improve, leading to the generation of increasingly realistic data [52].

WGAN improves the training stability of traditional GAN by replacing commonly used divergence metrics, which may not be continuous concerning the generator’s parameters, with the Earth-Mover (Wasserstein-1) distance. This distance measures the minimum cost of transforming one probability distribution into another by considering both the amount of mass moved and the transport distance. Under mild assumptions, the Wasserstein distance is continuous and differentiable almost everywhere, ensuring more stable training. However, the optimization process in WGAN can be challenging due to the interplay between the weight constraint and the cost function, which may result in vanishing or exploding gradients unless the clipping threshold is carefully tuned. To address these issues, WGAN with gradient penalty enforces the Lipschitz constraint using a gradient penalty, a more robust alternative to weight clipping.

A differentiable function is 1-Lipschitz if and only if its gradients have a norm of at most 1 everywhere. To enforce this constraint, WGAN with gradient penalty introduces a relaxed version by penalizing the gradient norm for random samples $\hat{x}∼P_{\hat{x}}$, ensuring stable training dynamics [53,54]. The new objective function is given as(8)$$L=E_{\tilde{x}∼P_{g}}\left[D\left(\tilde{x}\right)\right]-E_{x∼P_{r}}\left[D\left(x\right)\right]+\lambda \cdot E_{\hat{x}∼P_{\hat{x}}}\left[{\left(∥\nabla_{\hat{x}}D\left(\hat{x}\right){∥}_{2}-1\right)}^{2}\right],$$
where λ is the gradient penalty coefficient, $\hat{x}$ represents interpolated samples between real and generated data, $D(\hat{x})$ is the discriminator (critic) output, and $∥\nabla_{\hat{x}}D\left(\hat{x}\right){∥}_{2}$ is the L2 norm of the gradient of *D* with respect to $\hat{x}$. This penalty term ensures the gradient norm is close to 1, replacing the weight clipping approach in the original WGAN and providing smoother gradients for improved stability.

In the federated learning (FL) context, GANs have been employed with a single generator shared across all clients to generate consistent synthetic data. Each FL client trains a local discriminator on its data, and the updated models are aggregated at the server to build a global discriminator. However, training GANs in this semi-decentralized FL setup often yields inefficient results, as the generator struggles to learn from clients with diverse, non-IID data distributions, resulting in high generator loss. To address this inefficiency, the WGAN with gradient penalty is used by training the critic multiple times on local data and aggregating local discriminators into a global model [54].

In our implementation, the gradient penalty coefficient is set to 10 (λ=10). The critic is trained five times for each generator update to ensure an accurate approximation of the Wasserstein distance before updating the generator. This combination of gradient penalty and multiple critic updates significantly enhances gradient flow, reduces mode collapse, and ensures stable training dynamics, making WGAN with gradient penalty a robust alternative to traditional GANs in federated learning setups.

## 4. Experimental Evaluation

This section shows the experimental details of the proposed models. All experiments were conducted on Google Colab, which provided access to an NVIDIA Tesla T4 GPU with 12 GB of RAM through its cloud infrastructure. Both of the centralized and semi-decentralized FL-based models were implemented using the Python 3.10.12 programming language with TensorFlow 2.17.0, Keras 3.4.1, and scikit-learn 1.3.2. The clustering is implemented using PyClustering, which is an open-source library [55].

### 4.1. Datasets

#### 4.1.1. Training Dataset

The dataset used is CICIoT2023, which was created by the Canadian Institute for Cybersecurity (CIC) [56]. The CICIoT2023 dataset was generated using a topology that includes 105 IoT devices. Among these, 67 devices actively participated in the attacks, and five hubs connected an additional 38 Zigbee and Z-Wave devices. This setup replicates a real-world scenario of IoT products and services within a smart home environment. The devices used in this topology include smart home devices, cameras, sensors, and micro-controllers, all of which were interconnected and configured to facilitate various attacks and to capture the resulting attack traffic. Thus, the CICIoT2023 dataset is heterogeneous as it has a diversity of device types and communication protocols.

CICIoT2023 consists of 33 types of attacks that are divided into seven categories: distributed denial-of-service (DDoS), denial-of-service (DoS), Recon, web-based, Brute Force, Spoofing, and Mirai. The benign data are gathered from IoT traffic in idle states where the devices are configured with default parameters and without malicious or attacking scripts. The CICIoT2023 consists of 46,217,820 instances, each representing a network flow with 47 features. Figure 7 shows the number of instances for each attack category. We chose the CICIoT2023 dataset because it includes all classes in both the training and testing sets, unlike other datasets. Additionally, the number of instances per class is large enough to be distributed across the FL clients, allowing each client to be trained on each class. This, in turn, contributes to the development of an accurate global model. To the best of our knowledge, we are the first who use the whole CICIoT2023 dataset, considering all types of attacks, as other studies work with part of the dataset or consider specific types of ttack [57,58,59].

First of all, the dataset needs to be preprocessed and divided into training, validation, and testing sets with proportions of 60%, 20%, and 20%, respectively. The features are normalized using the StandardScaler method, while the classes (labels) are encoded using One-Hot Encoding for multi-class classification (34/8 classes) and Binary Encoding for binary classification.

#### 4.1.2. Testing Dataset

Three datasets are used for testing the proposed pre-trained model, namely BoT-IoT [60], WUSTL-IIOT-2021 [61], and Edge-IIoTset [62]. Our goal was to test the proposed model’s generalizability on additional datasets and improve its performance through fine-tuning. The reason for choosing those datasets is because they have similar features to the training dataset. Moreover, the DoS and DDoS attacks are presented in all the datasets, which are the main attacks that our model is targeting. More importantly, all datasets are designed for IoT traffic.

The BoT-IoT dataset was created by designing a realistic IoT network environment with five distinct IoT scenarios: a weather station, a smart fridge, remotely activated lights, motion-activated lights, and a smart thermostat. We utilized a 5% version extracted from the original dataset that has 35 features. The dataset includes 10 types of attacks: DDoS (HTTP, TCP, UDP), DoS (HTTP, TCP, UDP), OS Fingerprinting, Server Scanning, Keylogging, and Data Exfiltration attacks, as shown in Table 1. In total, 70% of the data are used for training, and the remaining 30% are used for testing. The original model had 47 input features, whereas the BoT-IoT datasets had 35 features. Thus, we modified the input layer of the model to accommodate this difference.

The WUSTL-IIOT-2021 dataset includes network traffic from Industrial Internet of Things (IIoT) systems that were used in cybersecurity research. The dataset has 41 features after removing 4 features, as the publisher of the dataset mentioned that “they are unique to the attacks and would expose the type of the attack to the model; therefore, the model would not be generalized for unseen data” [61]. In addition, the unused columns are dropped, which are ’StartTime’, ’LastTime’, ’SrcAddr’, ’DstAddr’, ’sIpId’, and ’dIpId’. Table 2 presents the dataset information. The dataset includes four types of different attacks: DoS, command injection, Reconnaissance, and Backdoor attacks. As DoS attacks typically generate high volumes of traffic and a large number of samples, 90% of the attack data are allocated to represent them. Other attack types occur less frequently, and when they do, they transmit only a limited amount of traffic data. Table 3 illustrates the statistics of the dataset. Overall, 80% is used for training, and the remaining 20% is used for testing.

The Edge-IIoTset dataset [62] was generated using a testbed that accurately simulates a real-world IoT/IIoT environment. The dataset includes real-world data, collected by executing realistic cyberattacks and capturing both legitimate and malicious network traffic. The dataset has 61 features and 14 types of attacks, which are divided into five categories: DDoS, injection, MITM, malware, and scanning attacks. Table 4 illustrates the class distributions in Edge-IIoTset. The dataset consists of several files, including attack traffic, normal traffic, and selected datasets for ML and DL, which contain two CSV files: DNN-EdgeIIoT-dataset.csv and ML-EdgeIIoT-dataset.csv. We use DNN-EdgeIIoT-dataset.csv to evaluate our model. Seventy percent of the data are used for training, while the remaining thirty percent are reserved for testing. We apply fine-tuning to all datasets to test our pre-trained model, as the input features differ from the pre-trained model and the number of classes varies. During fine-tuning, all layers are frozen except the input and output layers to retain the learned representations while allowing the model to adjust to the new feature space. This process significantly improves the model’s performance on the unseen datasets, demonstrating the efficacy of fine-tuning in cross-dataset generalization.

### 4.2. Performance Metrics

In this study, several performance evaluation indicators are employed to assess the effectiveness of the model. The following metrics are typically used in analyzing the intrusion detection performance:True Positive (TP): This represents the number of attack samples that are correctly classified as attacks.False Positive (FP): This refers to the number of benign samples that are incorrectly classified as attacks.True Negative (TN): This indicates the number of benign samples that are correctly classified as benign.False Negative (FN): This represents the number of attack samples that are incorrectly classified as benign.Accuracy: This metric indicates the proportion of correctly classified instances out of the total number of examples. It is calculated using(9)$$\mathrm{Accuracy}=\frac{TP+TN}{TP+FP+TN+FN}$$Precision: Precision measures the ratio of true positive predictions to the total number of positive predictions (both true and false). It is determined as(10)$$\mathrm{Precision}=\frac{TP}{TP+FP}$$Recall: Also known as sensitivity, recall quantifies the number of true positive predictions made out of all actual positives. It is given by(11)$$\mathrm{Recall}=\frac{TP}{TP+FN}$$F1-Score: This metric provides a balance between precision and recall by calculating their harmonic mean. The F1-score is computed using(12)$$\mathrm{F1-Score}=2\times \frac{\mathrm{Precision}\times \mathrm{Recall}}{\mathrm{Precision}+\mathrm{Recall}}$$

For multi-classifications, macro-averaging is used to calculate the recall, precision, and F1-score across all classes. As the dataset is imbalanced, the macro-averaging ensures that each class contributes equally to the evaluation by independently calculating and averaging the metrics for each class. Consequently, the overall performance measure is not biased toward any particular class. All performance metrics were computed using the scikit-learn (sklearn) library in Python. This library provides standardized implementations of these metrics, ensuring consistency in model evaluation.

### 4.3. Parameter Tuning

#### 4.3.1. Local Model Parameters

As BiLSTM has higher performance metrics, which will be shown in the performance evaluation section in Section 4.4, it is chosen as the local model for our proposed semi-decentralized FL model. The performance of any deep neural network is greatly affected by its hyper-parameters. Specifically, the BiLSTM model is influenced by the learning rate, batch size, dropout rate, activation function, number of layers, and number of neurons per layer. Thus, in this section, those hyper-parameters are tuned to make the model reach the best results. Since our model is targeting IoT environments with resource-constrained capabilities, the number of layers and the number of neurons per layer are tuned to provide a balance between performance and lightweight-ness, where the lightweight-ness is measured by the model size. Each parameter configuration is applied on each run, the results are averaged, and the standard deviation is calculated as presented in Table 5 and Table 6. After conducting a paired *t*-test to compare the performance metrics across different model parameters (m1, m2, m3, and m4), the results presented in Table 7 indicate no statistically significant difference in performance between models m1, m2, and m3. However, a significant difference was observed between m2 and m4, with m2 demonstrating superior performance. Given our primary objective of achieving high performance while minimizing model complexity, we selected m2. This configuration, with only two layers, provides a lightweight structure suitable for resource-constrained IoT devices while maintaining competitive performance.

After the parameter tuning, the BiLSTM model is configured with two layers (128 and 64 neurons), knowing that the input layer has the same number of features as CICIoT2023, which is 46, and the output layer has the same number of classes: 34, 8, and binary classes. The other hyper-parameters were configured after extensive experiments as follows: “ReLU” was used as the activation function in the hidden layers, with “Softmax” applied to the output layer for both the 34-class and 8-class models. However, for binary classification, a “Sigmoid” function was used in the output layer. To prevent overfitting, a dropout layer with a rate of 20% was added after each hidden layer, randomly ignoring 20% of neurons during training. The loss function used was categorical cross-entropy for multi-class classification and binary cross-entropy for binary classification. All experiments were conducted for 1 epoch with a batch size of 64 and a learning rate of 0.001, and the optimizer used was “adam”.

#### 4.3.2. FL Parameters

For the federated learning (FL) settings, the number of participating clients is set to 10, with the training set divided among them, ensuring that each client has samples from all types of attacks, as shown in Algorithm 1. Since the distribution is performed randomly, we generate five different client distributions, or “runs”. All experiments are performed on these five runs, and the results are averaged to minimize the impact of randomness. We set the number of FL rounds to 10 and use FedAvg as the aggregation algorithm.

#### 4.3.3. Clustering Algorithm

As mentioned above, clustering is performed using k-means. The number of clusters is set to 3 (k=3) based on the elbow method. Figure 8 demonstrates the elbow method to identify the optimal number of clusters. The Within-Cluster Error (WCE) decreases sharply up to k=3, after which the improvement slows, indicating k=3 as the optimal number of clusters.

K-means clustering is based on distance measures; thus, it needs to find the best distance measure that fits in our case. According to [63,64], we choose four distance metrics: Euclidean, Average Euclidean, Manhattan, and Coefficient of Divergence. Euclidean is chosen as it is the default for k-means clustering, despite its sensitivity to outliers. Therefore, Average Euclidean is used to manage outliers, while the Manhattan distance measure exhibits minimal distortion. On the other hand, the Coefficient of Divergence proves to be the most effective method for handling high-dimensional data. K-means clustering is applied using each one of the four distance measures; then, results are compared using the Silhouette score to choose the best distance measure. Since k-means clustering is a random algorithm where each run will give a new cluster, k-means clustering is applied 10 times, and then the average Silhouette score is calculated to compare between each one of the distance measures. According to the experimental results, the k-means with Manhattan distance achieves the highest Silhouette score. Table 8 below shows the Silhouette scores of applying k-means clustering with different distance measures.

### 4.4. Performance Evaluation

This section presents the evaluation results of the proposed model. First, it compares the semi-decentralized FL approach with the centralized FL approach. Next, the semi-decentralized FL approach is evaluated using three DL techniques: LSTM, BiLSTM, and WGAN.

#### 4.4.1. Centralized vs. Semi-Decentralized FL Using LSTM

To demonstrate the effectiveness of the semi-decentralized FL, we compare it with the centralized FL. Both approaches are configured with LSTM as a local model. The two approaches are applied on the five runs, and then the results are averaged. According to the results, the semi-decentralized FL-based model outperforms the centralized FL-based model in 34 and 8 classes, as shown in Figure 9. However, the centralized FL-based model has higher performance measures for the binary classification. This is because, in binary classification, the class imbalance is less than in multi-classification. This also demonstrates that the semi-decentralized approach is more effective at handling class imbalance than centralized FL.

Moreover, the semi-decentralized FL-based model reduces communication overhead by enabling the cluster head to communicate with the server, which results in preserving resource consumption (bandwidth) as well as improving the waiting time. The proposed clustering approach selects the cluster head based on the average distance to neighboring nodes within the cluster, aiming to efficiently receive weights and parameters without delay.

The resource consumption can be measured by the training time per FL round. For the centralized FL approach, the training time for one round is 4398.4525 s, and 3399.985 s for the semi-decentralized FL approach. Therefore, the semi-decentralized FL approach contributes to the preservation of approximately 1000 s of training time per FL round, thereby maintaining the total training time at the end of the FL learning process. This is due to the clustering mechanism, which allows each cluster to send its updates to the server without waiting for other clusters. As a result, training time is minimized, which indeed helps in reducing the resource consumption. Figure 10 shows the comparison of the training time per FL round for centralized vs. semi-decentralized approaches, knowing that both models are converged at round 4.

#### 4.4.2. Semi-Decentralized FL with Different DL Models

Since the semi-decentralized FL outperforms centralized FL, this section will focus on improving the performance of the semi-decentralized FL architecture by trying other DL models. As mentioned earlier, there are five different client distributions known as runs. The FL process is applied to all the runs, the results are averaged, and the standard deviation is calculated. The model is applied to the 34-class, 8-class, and binary classification tasks. Three deep learning (DL) models are used: LSTM, BiLSTM, and WGAN, as described in Section 3.4. Table 9 and Figure 11 below show the results of applying the three DL models: LSTM, BiLSTM, and WGAN as local models for the semi-decentralized FL approach. The results indicate that BiLSTM has the highest performance measures.

When a paired *t*-test is applied to study if the difference is statistically significant, the results indicate that there is a significant difference in performance between all models in all metrics except precision. Table 10 shows the results of the application of the paired *t*-test.

### 4.5. Performance Comparisons

The pre-trained semi-decentralized FL-based model with BiLSTM is tested on three datasets: BoT-IoT, WUSTL-IIoT-2021, and Edge-IIoTset.

As mentioned previously, all datasets have different input features as well as numbers of classes. Thus, fine-tuning is applied to the pre-trained model where all layers are frozen except the input and output layers to retain the learned representations while allowing the model to adjust to the new feature space. This process significantly improves the model’s performance on the unseen datasets, demonstrating the efficacy of fine-tuning in cross-dataset generalization. As there are five runs for the proposed semi-decentralized FL model, there will be five pre-trained models that can be used for testing an unseen dataset. For each model, we calculate the average F1-score, as it is the harmonic mean of precision and recall, and choose the model that has the closest value to this average. Since there are three classifications per model (34-class, 8-class, and binary), one model is chosen for each classification based on the average F1-score.

#### 4.5.1. BoT-IoT Testing Results

The number of input features in BoT-IoT is 35; however, the pre-trained model has 46 features, as well as a different number of classes. Therefore, the input and output layers are adjusted to match the number of features and classes of the BoT-IoT dataset, while the hidden layers are frozen. The semi-decentralized FL-based model with BiLSTM (34 classes, as it includes the different types of DoS and DDoS attacks as in BoT-IoT), as a local model, is tested on the BoT-IoT test set (which is 30% of the dataset). Then, we compare our results with the results obtained from [22], who proposed a multi-attack detection mechanism called LocKedge (Low-Complexity Cyberattack Detection in IoT Edge Computing) that uses the BoT-IoT dataset. The proposed model, Lockedge, relies on a centralized FL approach. Table 11 demonstrates the detection rate (recall) for each type of attack.

Compared with the Lockedge model, the semi-decentralized FL-based model improves the detection rate in classes, while in some classes, the detection rate is not improved. Our proposed model achieves the highest detection rate for DDoS attacks, OS Fingerprint, and Server Scanning attacks. However, for Keylogging and Data Theft attacks, the detection rate is 0, which comes as a result of having a low number of samples compared with other attack categories. As presented in Table 1, there is a significant imbalance in the number of samples for each class, making the distribution of these samples among FL clients a challenging task. Even when reducing the number of clients, the client still needs enough samples per class to train its local model, which in turn will aid in generating an accurate global model.

#### 4.5.2. WUSTL-IIoT-2021 Testing Results

The WUSTL-IIoT-2021 dataset has 41 features and five classes: DoS, command injection, Reconnaissance, Backdoor attacks, and normal. As the number of input features and number of classes vary from the pre-trained model, fine-tuning is applied to the input and output layers. Then, our proposed model with three classes is tested on the testing set (which is 20% of the dataset). Eid et al. [65] proposed an IDS for IIoT that utilized WUSTL-IIoT-2021. The proposed IDS is based on an optimized CNN model, and it examines the impact of artificially balancing the training datasets on the CNN model’s performance. To facilitate this analysis, two types of training subsets were prepared: one that retained the natural imbalance of the datasets and another that was deliberately balanced to equalize the class distributions. A comparison of our testing results with the [65] results is demonstrated in Table 12, Table 13 and Table 14 below. We compare our results with the unbalanced since we did not apply any oversampling technique and kept the dataset unbalanced.

According to the results of testing on the WUSTL-IIoT-2021 dataset, our proposed semi-decentralized FL model with BiLSTM achieves the highest performance measures for most attack types. However, its performance is lower for normal and DoS traffic because these classes have the majority of samples in the dataset, as shown in Table 3. This high class imbalance results in lower performance.

Table 15 presents the average performance measures, demonstrating that the semi-decentralized FL model achieves the highest accuracy.

DeepIIoT is proposed by [66], which utilizes WUSTL-IIoT-2021 to train MLP to detect anomalies. They tackle the issue of class imbalance by randomly undersampling the benign class, which reduces the percentage of benign instances from 92% to 66%. This helps to improve the F1-score where the minority class is being falsely classified as the majority. We tested our proposed model in binary classification to compare it with the DeepIIoT model, as well as the model proposed in [65]. The results are presented in Table 16, which shows that our model (semi-decentralized FL with BiLSTM) achieves the highest results in accuracy and precision, knowing that the classes are kept imbalanced, resulting in lower results in other metrics.

#### 4.5.3. Edge-IIoTset Testing Results

Friha et al. [62] evaluated the Edge-IIoTset in both classical ML/DL (non-federated learning) and federated learning. They use DNN-EdgeIIoT-dataset.csv and split the data into 70% for training and the remaining 30% for testing. There are 93 features and three different classification types: 15, 6, and binary classifications. Therefore, the input and output layers are adjusted to match the number of features and classes of the Edge-IIoTset dataset while the hidden layers are frozen. We use the same test split to evaluate our semi-decentralized FL-based model with 34 classes because it has DDoS attack types as in Edge-IIoTset. Table 17 illustrates a comparison of our evaluation results for each type of attack with the classical ML/DL (non-federated learning).

The attack types in Table 17 are Backdoor attack (Back), HTTP flood DDoS attack (HTTP), ICMP flood DDoS attack (ICMP), TCP SYN Flood DDoS attack (TCP), UDP flood DDoS attack (UDP), OS Fingerprinting attack (Fing), man in the middle attack (MITM), password-cracking attack (Pwd), Port Scanning attack (Port), Ransomware attack (Rans), SQL Injection (SQL), Upload attack (Upload), vulnerability scanning attack (Scan) and Cross-site Scripting attack (XSS).

According to the results, the semi-decentralized FL model with BiLSTM has higher performance measures in some types of attacks. Specifically, the proposed model achieves the highest performance metrics in detecting HTTP flood DDoS, TCP SYN Flood DDoS, password-cracking, and SQL Injection attacks. It also has the highest precision and F1-score in detecting Port Scanning attacks.

The centralized FL-based model that was introduced by [62] is evaluated using the Edge-IIoTset. Thus, we compare our test results with their results considering the same FL parameters: 10 FL rounds, 10 FL clients, and the non-IID dataset. Their model attained an accuracy of 91.45%, which is lower than the accuracy of our proposed model that achieves 95.17%.

### 4.6. Discussion

This study proposed a semi-decentralized federated learning (FL) framework that leverages client clustering to address the heterogeneity and limited resource capabilities of IoT networks. The heterogeneity of IoT devices affects the performance of FedAvg, the aggregation algorithm used in this work, which requires having similar data patterns to perform better. Thus, clustering helps to improve the aggregation process, which, in turn, improves the performance of the global model. Moreover, clustering reduces the communication overhead as the server communicates with the cluster heads instead of each client. Specifically, the cluster head collects the weights from each client within the cluster and sends them to the server. Consequently, the semi-decentralized FL approach has demonstrated improvements in performance and resource consumption, while also addressing the data heterogeneity issue. In other clustered-based FL models, the cluster head conducts multiple rounds of aggregation to obtain the cluster model. Then, the server aggregates the received cluster models to obtain the global model. These two levels of aggregation are time- and resource-consuming. However, in our proposed model, the cluster head aggregates the clients’ model updates within the cluster and sends the aggregated model to the server. The server then aggregates the cluster’s models and returns the global model to the cluster heads, all within one FL round, thereby reducing the communication overhead and making it affordable for resource-constrained IoT.

The proposed model is configured with three DL techniques as local models, namely LSTM, BiLSTM, and WGAN, using the CICIoT2023 dataset. The WGAN is applied in the FL framework with one generator for all FL clients to train them on the same generated data. This is because our concern is to detect intrusion, not to know the real from the generated data. Then, each FL client has a local discriminator model to train it on its local data, sends the updated model to the global discriminator in the server for aggregation, and generates the global model. However, in other FL models that use GAN or WGAN, each FL client has a local generator/discriminator, and the server has a global generator/discriminator. This is because their main concern is to use the local generator to balance the classes in the local dataset in each client [37,67]. The experimental results demonstrate that the BiLSTM achieves the best performance, followed by the LSTM, whereas the WGAN shows the lowest performance. This is due to the heterogeneity issue inherent in FL that causes WGAN to struggle in generating meaningful data distributions that effectively represent all attack classes. Consequently, discriminators trained on these generated data exhibit reduced accuracy in detecting attacks.

Our primary objective is to design a lightweight IDS that fits the capabilities of IoT networks. To achieve this, we configure the BiLSTM, with a number of layers and neurons that strikes a balance between performance and lightweight-ness, with the model size serving as a gauge of lightweight-ness. After conducting extensive experiments, we found that the BiLSTM with two layers, each containing 128 and 64 neurons, yields the best results in terms of both performance and lightweight-ness.

Furthermore, we evaluate our proposed semi-decentralized FL-based model using an IoT dataset, while other semi-decentralized approaches rely on image datasets like CIFAR, MNIST, etc. Most of the proposed IDSs for IoT networks have utilized datasets that do not have IoT traces, such as the NSL-KDD, UNSW-NB15, and CICDDoS2019 datasets [65]. In our experiment, the CICIoT2023 is used with large IoT traffic, whereas many papers use part of the dataset to reduce the computation overhead [58,59]. Nevertheless, we cannot compare with their work as their models are trained on different data samples.

For the other FL approaches (both centralized and decentralized) that are being evaluated using IoT datasets, they did not consider distributing the dataset between clients, considering that each client has all types of attacks. Most of them distribute the dataset based on the IP address, as it is a source of attacks [22,68]. However, after investigating those datasets, it was found that the IP address of the attack does not contain all types of attacks. Thus, some FL clients would not be trained on these attacks, which results in low detection rates for those attacks.

Comprehensively, all the results demonstrate that our proposed semi-decentralized FL effectively produces a lightweight and heterogeneity-aware model for detecting intrusions in IoT networks. Its effectiveness is particularly notable in detecting DDoS attacks, highlighting its potential for real-world cybersecurity applications.

## 5. Conclusions and Future Work

In this paper, we have introduced a semi-decentralized FL-based model which is a lightweight and heterogeneity-aware intrusion detection mechanism that is suitable for deployment in IoT networks. Our proposed model is based on clustering FL clients, which solves the data heterogeneity, which, in turn, improves the learning process as well as reduces the communication overhead by working at the cluster level instead of the client level.

The proposed model is evaluated using the CICIoT2023 dataset as it has a large number of IoT traces and attack categories, particularly DDoS, as it is our target attack. Three DL techniques are being evaluated as local models in a semi-decentralized FL approach, namely LSTM, BiLSTM and WGAN. The results show that the BiLSTM achieves the best performance; thus, it is chosen as the local model in the proposed model. After extended experiments, the BiLSTM with two layers, each with 128 and 64 neurons, respectively, provides the best results in terms of performance, and lightweight-ness makes it affordable for resource-constrained IoT. The pre-trained semi-decentralized FL model was further tested on three additional IoT datasets—BoT-IoT, WUSTL-IIoT-2021, and Edge-IIoTset—to validate its generalizability. The results show that the proposed model achieves the highest performance measures in most classes, with exceptional performance in the detection of DDoS attacks.

In future work, we will improve the performance of the FL-based models and lighten the local model more to better suit IoT devices with low resources. Compression techniques such as weight pruning, quantization, and others can be used to lighten the IDS model. Since the aggregation algorithm plays a crucial role in the FL process, particularly in heterogeneous and resource-constrained IoT environments, this requires designing an aggregation algorithm specifically for this purpose. Finally, we aim to deploy the proposed IDS model in a real IoT network, where the FL server functions as the edge device, and the IoT device, or end node, acts as the FL client.

## Figures and Tables

**Figure 1 sensors-25-01039-f001:**
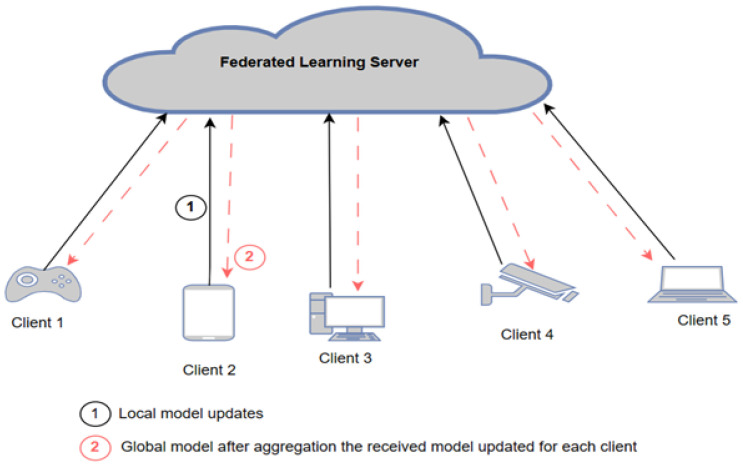
hlCentralized FL (client–server) architecture.

**Figure 2 sensors-25-01039-f002:**
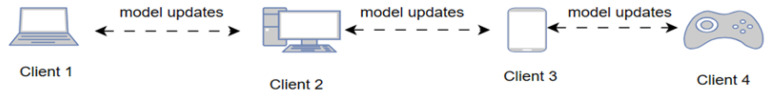
Decentralized FL (D2D) architecture.

**Figure 3 sensors-25-01039-f003:**
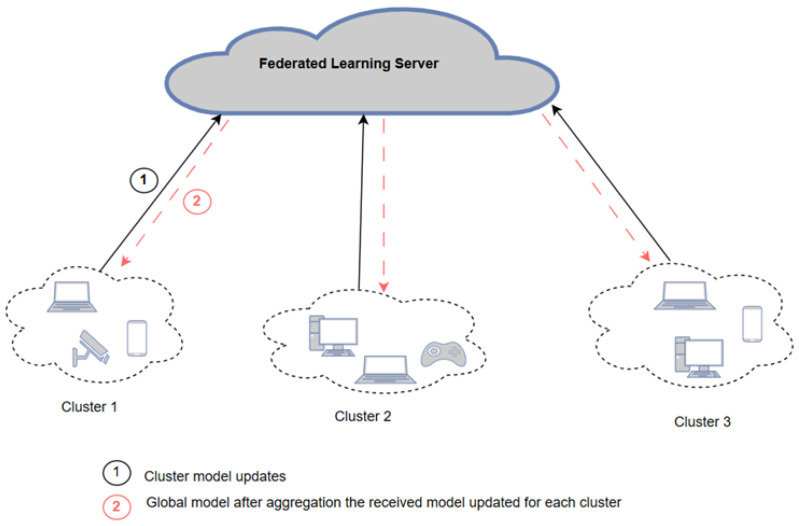
Semi-decentralized FL architecture.

**Figure 4 sensors-25-01039-f004:**
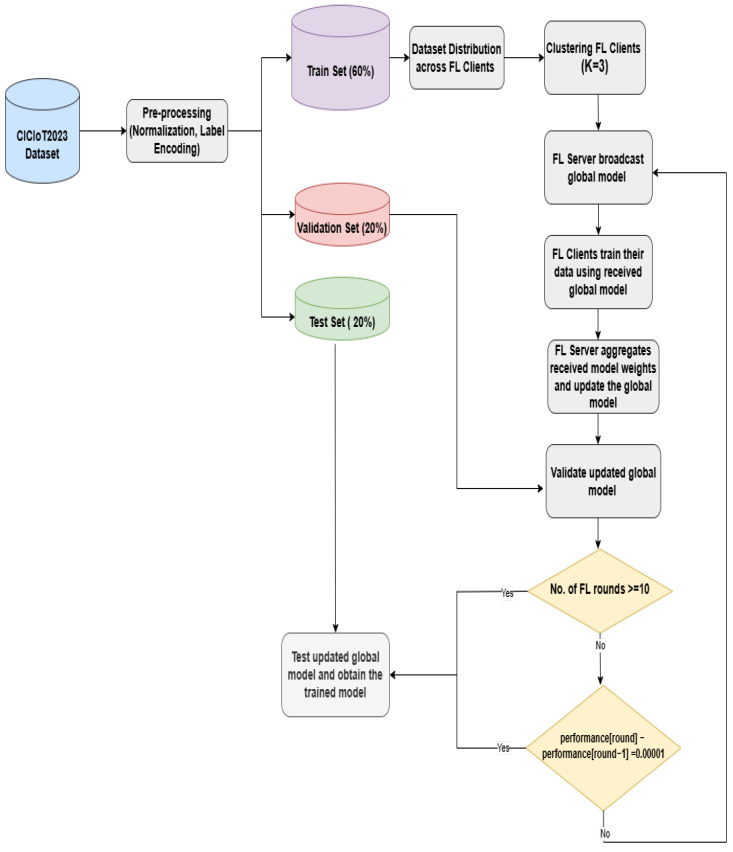
Illustration of semi-decentralized FL architecture.

**Figure 5 sensors-25-01039-f005:**
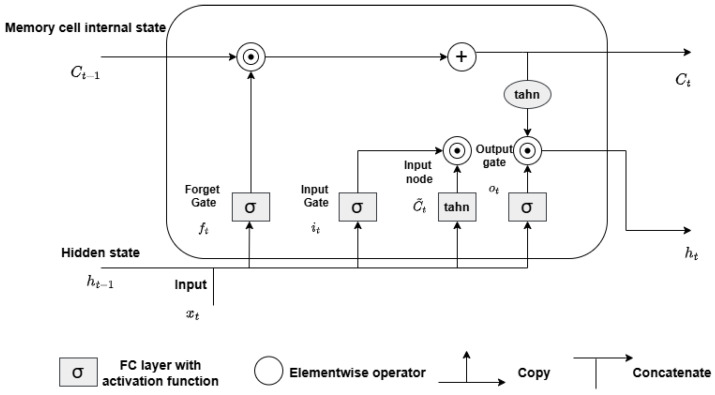
LSTM cell architecture.

**Figure 6 sensors-25-01039-f006:**
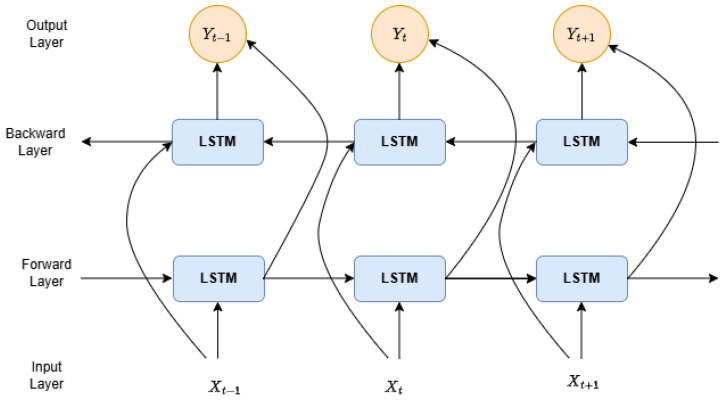
Illustration of bidirectional LSTM.

**Figure 7 sensors-25-01039-f007:**
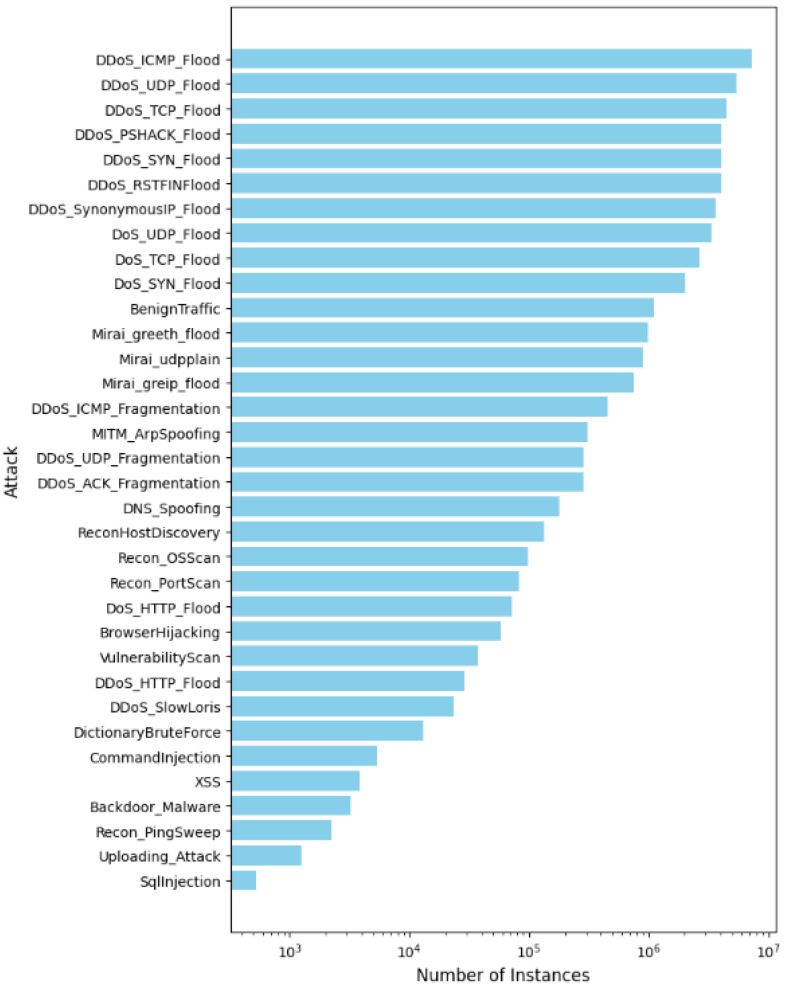
CICIoT2023 class distribution.

**Figure 8 sensors-25-01039-f008:**
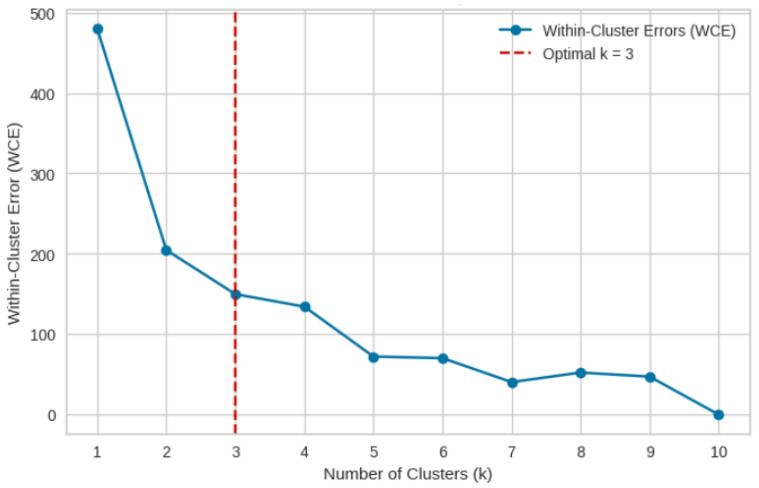
The elbow method plot showing the Within-Cluster Error (WCE) against the number of clusters (*k*). The optimal number of clusters is identified as k=3, where the WCE decreases sharply before leveling off.

**Figure 9 sensors-25-01039-f009:**
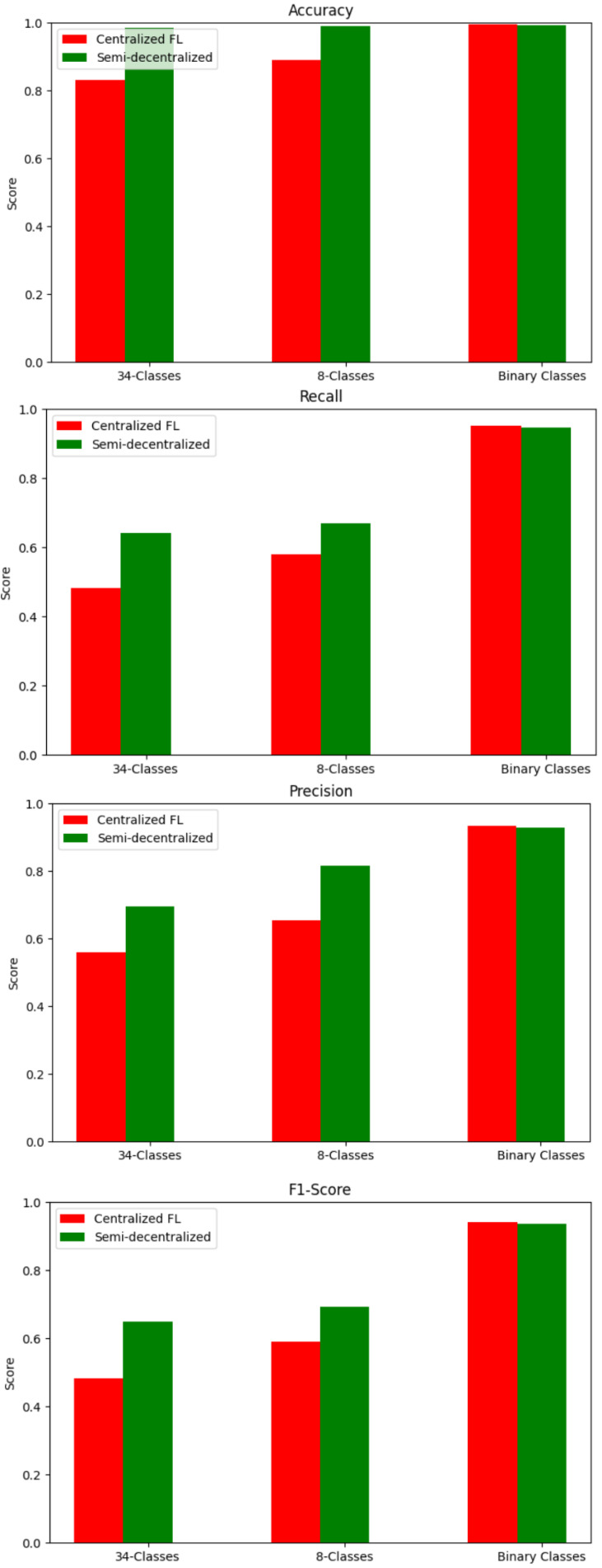
Performance comparison for centralized vs. semi-decentralized FL approach.

**Figure 10 sensors-25-01039-f010:**
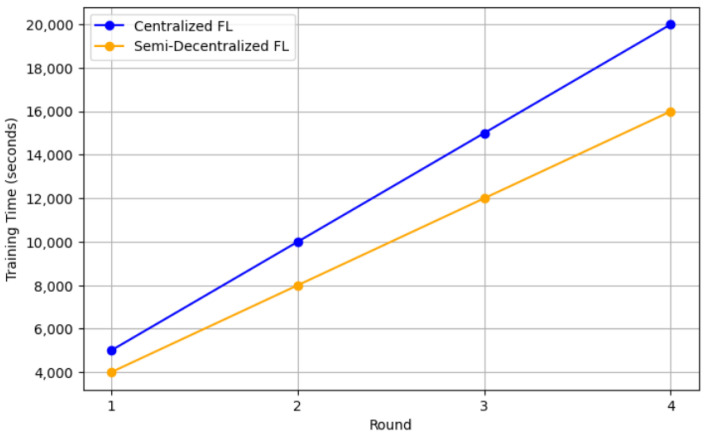
Training time per FL round for centralized vs. semi-decentralized approach.

**Figure 11 sensors-25-01039-f011:**
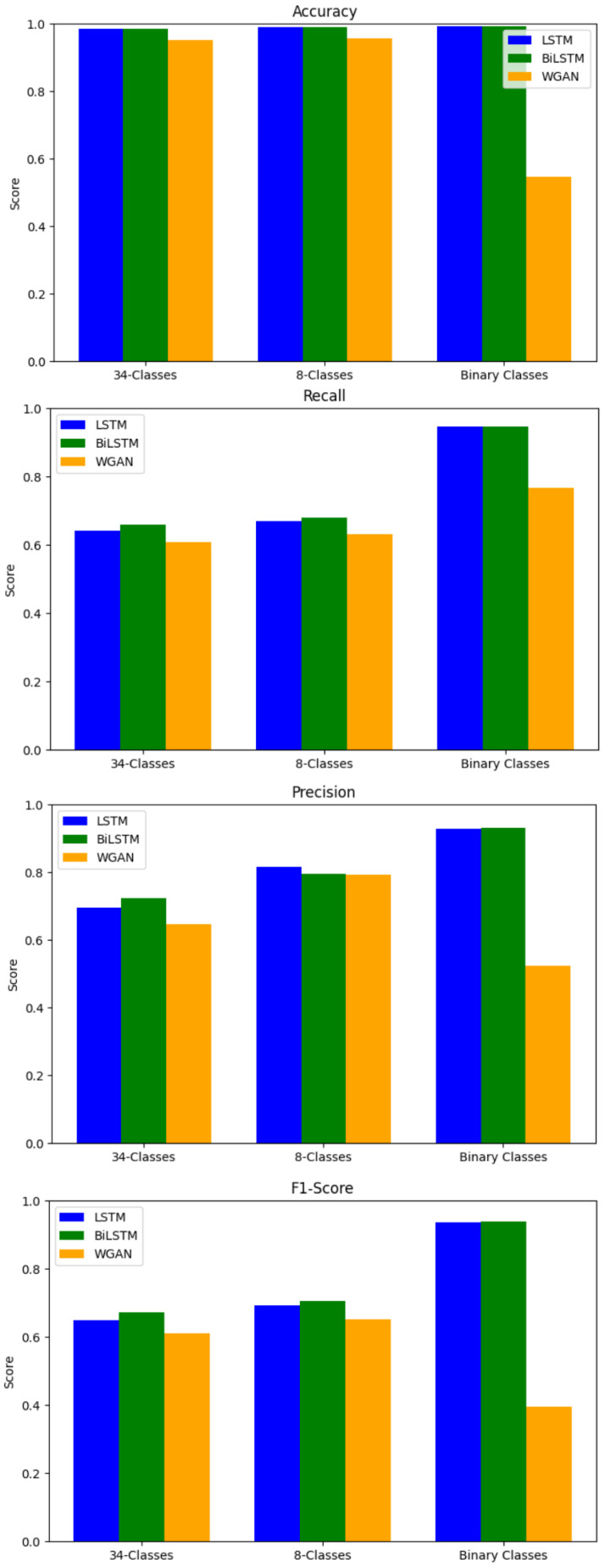
Comparison of semi-decentralized FL models (LSTM, BiLSTM, and WGAN) across different class categories.

**Table 1 sensors-25-01039-t001:** Types of attack and number of samples in BoT-IoT dataset.

Types of Attack	Number of Samples
DoS-HTTP	1485
DoS-TCP	615,800
DoS-UDP	1,032,975
DDoS-HTTP	989
DDoS-TCP	977,380
DDoS-UDP	948,255
OS Fingerprinting	17,914
Server Scanning	73,168
Keylogging	73
Data Theft	6
Normal	477
**Totals**	3,668,522

**Table 2 sensors-25-01039-t002:** WUSTL-IIOT-2021 dataset specification.

**Number of Samples**	1,194,464
**Number of features**	41
**Number of attack samples**	87,016
**Number of normal samples**	1,107,448

**Table 3 sensors-25-01039-t003:** Statistical information of the traffic types in WUSTL-IIOT-2021.

Traffic Type	Percentage (%)
Normal Traffic	92.72
Total Attack Traffic	7.28
Command Injection Traffic	0.31
DoS Traffic	89.98
Reconnaissance Traffic	9.46
Backdoor Traffic	0.25

**Table 4 sensors-25-01039-t004:** Types of attack and number of samples in Edge-IIoTset.

Attack Category	Attacks	No. of Instances
**DoS/DDoS Attacks **	DDoS_HTTP attack	229,022
	DDoS_ICMP attack	2,914,354
	DDoS_TCP attack	2,020,120
	DDoS_UDP attack	3,201,626
**Injection Attacks**	SQL_injection attack	51,203
	Uploading attack	37,634
	XSS attack	15,915
**Malware Attacks**	Backdoor attack	24,862
	Password attack	1,053,385
	Ransomware attack	10,925
**Scanning attacks**	Port_Scanning attack	22,564
	Vulnerability_scanner attack	145,869
	Fingerprinting attack	1001
**MITM attack**		1229
**Normal**		11,223,940
**Total**		20,952,648

**Table 5 sensors-25-01039-t005:** Average performance measures for each parameter configuration in BiLSTM.

Parameters	Avg. Accuracy	Avg. Recall	Avg. Precision	Avg. F1-Score
m1: 1 layer (64 neurons)	0.99	0.6741	0.7567	0.6935
m2: 2 layers (128, 64)	**0.9909**	**0.6805**	0.7948	**0.7045**
m3: 3 layers (128, 64, 32)	0.9896	0.6692	**0.8143**	0.6916
m4: 4 layers (128, 64, 64, 32)	0.9859	0.6664	0.7849	0.6861

**Table 6 sensors-25-01039-t006:** Standard deviation of performance measures for each parameter configuration in BiLSTM.

Parameters	St. Dev Accuracy	St. Dev Recall	St. Dev Precision	St. Dev F1-Score
m1: 1 layer (64 neurons)	0.0004	0.0026	0.0098	0.0021
m2: 2 layers (128, 64)	0.0001	0.0025	0.0064	0.0028
m3: 3 layers (128, 64, 32)	0.0014	0.0011	0.0196	0.0025
m4: 4 layers (128, 64, 64, 32)	0.0079	0.0017	0.0178	0.0034

**Table 7 sensors-25-01039-t007:** Paired *t*-test results for performance comparison between different parameter configurations in BiLSTM.

Parameters	*p*-Value	Significance
m1 vs. m2	0.1864	Not Significant
m1 vs. m3	0.4634	Not Significant
m1 vs. m4	0.8124	Not Significant
m2 vs. m3	0.8534	Not Significant
m2 vs. m4	0.0257	Significant
m3 vs. m4	0.2029	Not Significant

**Table 8 sensors-25-01039-t008:** Silhouette scores for different distance measures.

Distance Measure	Silhouette Score
Euclidean	0.2654
Average Euclidean	0.1823
Manhattan	**0.2777**
Coefficient of Divergence	0.14398

**Table 9 sensors-25-01039-t009:** Comparison of classification metrics for semi-decentralized FL models (LSTM, BiLSTM, WGAN).

Metrics	Classes	Semi-Dec FL LSTM	Semi-Dec FL BiLSTM	Semi-Dec FL WGAN
Accuracy	34-Classes	0.9843	**0.9855**	0.9516
	8-Classes	0.9902	**0.9909**	0.9569
	Binary classes	0.9942	**0.9943**	0.5464
Recall	34-Classes	0.6409	**0.6602**	0.6095
	8-Classes	0.6701	**0.6805**	0.6309
	Binary classes	0.9467	**0.9474**	0.7673
Precision	34-Classes	0.6959	**0.7227**	0.6474
	8-Classes	0.8158	**0.7948**	0.7927
	Binary classes	0.9293	**0.9307**	0.5246
F1-Score	34-Classes	0.6492	**0.6731**	0.6099
	8-Classes	0.6931	**0.7054**	0.6511
	Binary classes	0.9378	**0.9389**	0.3957

**Table 10 sensors-25-01039-t010:** Results of paired *t*-test of all metrics values on the proposed model with three DL techniques (BiLSTM, LSTM, WGAN).

Metric	Comparison	*p*-Value	Significance
Accuracy	BiLSTM vs. LSTM	0.0054	Significant
	BiLSTM vs. WGAN	0.0443	Significant
	LSTM vs. WGAN	0.0470	Significant
Recall	BiLSTM vs. LSTM	0.0004	Significant
	BiLSTM vs. WGAN	0.0047	Significant
	LSTM vs. WGAN	0.0097	Significant
Precision	BiLSTM vs. LSTM	0.0562	Not Significant
	BiLSTM vs. WGAN	0.8959	Not Significant
	LSTM vs. WGAN	0.1430	Not Significant
F1-Score	BiLSTM vs. LSTM	0.0019	Significant
	BiLSTM vs. WGAN	0.0017	Significant
	LSTM vs. WGAN	0.0032	Significant

**Table 11 sensors-25-01039-t011:** Detection rate (recall) comparison across different attack types using various models.

Attack Type	KNN	DT	RF	SVM	LocKedge	Proposed Semi-Decentralized-FL BiLSTM
DoS-HTTP	0.81690	0.84507	0.76056	0.74647	0.90862	0.6879
DoS-TCP	**1.0000**	0.99752	**1.0000**	**1.0000**	**1.0000**	0.9647
DoS-UDP	0.99851	0.99926	0.99926	0.99554	0.99928	0.7292
DDoS-HTTP	0.96774	0.82258	0.96774	0.97581	0.98715	**0.9998**
DDoS-TCP	0.99173	0.97746	0.99248	0.99624	0.99965	**1.0000**
DDoS-UDP	0.99217	**1.0000**	**1.0000**	0.96784	**1.0000**	**1.0000**
OS Fingerprinting	0.93478	0.93478	0.89130	0.78261	0.99258	**1.0000**
Server Scanning	0.97826	**1.0000**	**1.0000**	0.98913	0.99973	**1.0000**
Keylogging	**1.0000**	0.30000	0.90000	**1.0000**	0.99268	0.0000
Data Theft	**1.0000**	0.00000	0.00000	0.00000	0.56098	0.0000

**Table 12 sensors-25-01039-t012:** Precision comparison of testing WUSTL-IIoT-2021 for multi-classification.

Attack Scenario	Unbalanced (CNN)	Proposed Semi-Decentralized-FL BiLSTM
Backdoor	0.7685	**0.9952**
Command injection	0.9731	**0.9991**
Denial of service	0.9990	**1.0000**
Reconnaissance	0.9971	**0.9997**
Normal	**0.9995**	0.7188

**Table 13 sensors-25-01039-t013:** Recall comparison of testing WUSTL-IIoT-2021 for multi-classification.

Attack Scenario	Unbalanced (CNN)	Proposed Semi-Decentralized-FL BiLSTM
Backdoor	0.7830	**0.9979**
Command injection	0.8378	**0.9966**
Denial of service	**0.9939**	0.7045
Reconnaissance	0.9999	**0.9999**
Normal	**0.9999**	0.6053

**Table 14 sensors-25-01039-t014:** F1-score comparison of testing WUSTL-IIoT-2021 for multi-classification.

Attack Scenario	Unbalanced (CNN)	Proposed Semi-Decentralized-FL BiLSTM
Backdoor	0.7757	**0.9966**
Command injection	0.9004	**0.9978**
Denial of service	**0.9965**	0.8267
Reconnaissance	0.9985	**0.9998**
Normal	**0.9997**	0.6571

**Table 15 sensors-25-01039-t015:** Comparison of WUSTL-IIoT-2021 testing results.

Metric	Unbalanced CNN	Proposed Semi-Decentralized-FL BiLSTM
Accuracy	0.9994	**0.9996**
Precision	**0.9474**	0.9425
Recall	**0.9229**	0.8608
F1-score	**0.9341**	0.8956

**Table 16 sensors-25-01039-t016:** WUSTL-IIoT-2021 binary testing results.

Model	Accuracy	Precision	Recall	F1-Score
DeepIIoT	0.9994	0.9992	**0.9995**	**0.9994**
Optimized CNN	0.9982	0.9469	0.9284	0.9377
Semi-decentralized FL (BiLSTM)	**0.9996**	**0.9995**	0.9979	0.9987

**Table 17 sensors-25-01039-t017:** Performance comparison of different ML/DL algorithms (non-federated) across various attack types (Pr: precision; Re: recall; F1: F1-score; Metr: metrics).

Alg	Metr	Normal	Back	HTTP	ICMP	TCP	UDP	Fing	MITM	Pwd	Port	Rans	SQL	Upload	Scan	XSS
DT	Pr	**1.00**	0.86	0.44	**1.00**	0.66	**1.00**	0.00	0.00	0.00	0.71	0.80	0.34	**1.00**	**1.00**	0.15
	Rc	**1.00**	0.78	0.63	**1.00**	**1.00**	**1.00**	0.00	**1.00**	0.00	0.48	0.86	0.96	0.02	**1.00**	0.23
	F1	**1.00**	0.82	0.52	**1.00**	0.80	**1.00**	0.00	0.00	0.00	0.58	0.83	0.50	0.04	**1.00**	0.18
RF	Pr	**1.00**	0.99	0.64	0.96	0.82	**1.00**	0.77	**1.00**	0.41	0.63	0.96	0.76	0.66	**1.00**	0.65
	Rc	**1.00**	0.92	0.82	**1.00**	0.75	**1.00**	0.10	**1.00**	0.81	**1.00**	0.93	0.21	0.51	0.81	0.58
	F1	**1.00**	0.95	0.72	0.98	0.77	**1.00**	0.18	**1.00**	0.77	0.77	0.95	0.33	0.58	0.90	0.61
Knn	Pr	**1.00**	0.96	0.69	**1.00**	0.76	**1.00**	0.79	0.97	0.45	0.74	0.95	0.74	0.63	**1.00**	0.49
	Rc	**1.00**	0.94	0.74	**1.00**	0.76	**1.00**	0.10	**1.00**	0.56	0.73	0.94	0.49	0.55	0.88	0.57
	F1	**1.00**	0.95	0.72	**1.00**	0.78	**1.00**	0.74	0.99	0.50	0.73	0.95	0.55	0.55	0.90	0.53
SVM	Pr	**1.00**	0.63	0.86	**1.00**	0.74	**1.00**	0.80	**1.00**	0.60	0.64	0.69	0.42	0.66	0.95	0.61
	Rc	**1.00**	0.77	0.60	**1.00**	0.59	**1.00**	0.66	**1.00**	0.23	**1.00**	0.51	0.82	0.40	0.86	0.88
	F1	**1.00**	0.69	0.71	0.99	0.71	1.00	0.72	**1.00**	0.34	0.78	0.59	0.55	0.90	0.86	0.72
DNN	Pr	**1.00**	0.95	0.76	**1.00**	0.82	**1.00**	0.61	**1.00**	0.45	0.66	0.79	0.57	0.56	0.90	0.43
	Rc	**1.00**	0.86	0.92	**1.00**	0.90	**1.00**	0.64	**1.00**	0.38	0.50	0.85	0.71	0.48	0.85	0.37
	F1	**1.00**	0.90	0.83	**1.00**	0.90	**1.00**	0.61	**1.00**	0.45	0.66	0.79	0.57	0.56	0.90	0.43
Semi-dec FL(BiLSTM)	Pr	0.9907	0.9491	**1.00**	**1.00**	**1.00**	0.9407	0.4710	0.9462	**0.9986**	**0.9982**	0.6774	0.8565	0.4488	**1.00**	0.8344
	Rc	0.9802	0.5914	**1.00**	0.9986	**1.00**	0.8305	0.7653	0.8456	**1.00**	0.9590	0.5394	0.9965	0.8875	0.5932	0.1901
	F1	0.9854	0.7288	**1.00**	0.9993	**1.00**	0.8822	0.5831	0.8931	**0.9993**	**0.9782**	0.6006	0.9212	0.5962	0.7446	0.3097

## Data Availability

The dataset we used is publicly available and is titled “CICIoT2023”, provided by the Canadian Institute for Cybersecurity (CIC) at UNB. You can find it under their Research Datasets section.

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
