# Peer review of "A Heterogeneity-Aware Semi-Decentralized Model for a Lightweight Intrusion Detection System for IoT Networks Based on Federated Learning and BiLSTM"

_sensors, 2025, doi:10.3390/s25041039_

Round 1

Reviewer 1 Report

Comments and Suggestions for Authors

The paper presents a semi-decentralized federated learning (FL) model for lightweight intrusion detection systems (IDS) in IoT networks, leveraging BiLSTM and WGAN as local models. While the study addresses an important and timely problem—securing resource-constrained IoT devices—the paper has several  weaknesses that impact its overall quality and contribution.

1、Ablation studies are missing, making it difficult to assess the contribution of individual components (e.g., clustering, BiLSTM, WGAN) to the overall performance. No sensitivity analysis is performed to evaluate the impact of key parameters (e.g., number of clusters, FL rounds).

2、The paper does not compare the proposed model with state-of-the-art FL-based IDS approaches, such as those using GANs or blockchain.The comparison is limited to detection rate and does not include computational efficiency metrics (e.g., training time, memory usage).

3、The claim that the model is lightweight is not sufficiently validated with metrics like memory usage, energy consumption, or inference time.The paper does not adequately discuss the limitations of the proposed model, such as its performance on rare attack types (e.g., Keylogging, Data Theft) or scenarios with extreme heterogeneity.

4、No experiments are conducted to assess the impact of clustering on the model’s performance (e.g., comparing clustered vs. non-clustered FL).The paper does not explore the trade-offs between model performance and lightweight-ness (e.g., reducing the number of layers or neurons in BiLSTM).

5、The statistical significance of the results has not been tested, making it difficult to evaluate the robustness of the results. And some indicators of the experimental structure in the paper have obvious weaknesses compared to existing methods, but the paper did not fully explain the reasons or rationality of their existence.

6、The WGAN parameters (e.g., number of critic updates, gradient penalty) are not discussed in detail, making it difficult to understand the training process. The clustering parameters (e.g., number of clusters, k=3) are chosen based on the Elbow method, but the paper does not show the Elbow plot or justify the choice of k=3. No sensitivity analysis is performed to evaluate the impact of key parameters (e.g., number of clusters, FL rounds) on the model’s performance. Comments on the Quality of English Language
  • Some sections, particularly the discussion of FL architectures, are repetitive and could be more concise.

  • Some references are outdated or not directly relevant (e.g., [2], [3] on energy-based IDS, which are less related to the core FL contribution).

  • The paper does not sufficiently compare its work with recent FL-based IDS approaches, such as those using GANs or blockchain for IoT security.

  • Missing references to state-of-the-art clustering techniques in FL (e.g., hierarchical clustering, DBSCAN) that could have been considered.

Author Response

Thank you very much for taking the time to review this manuscript, we really appreciate your valuable comments. Please find the detailed responses in the attached file.

Reviewer 2 Report

Comments and Suggestions for Authors

The authors propose a semi-decentralized Federated Learning-based model for a lightweight Intrusion detection System. The idea is interesting, and the paper is well written; however, the following comments need to be addressed: 

1/The authors' names should be explicitly mentioned in Sections 2.1 and 2.2 for better clarity of previous work.  

2/The quality of the figures is not acceptable. All figures need  improvement to enhance readability and comprehension.  

3/The motivation for using BiLSTM and WGAN models is unclear. Please elaborate on why these models were chosen and how they align with the problem being addressed.  

4/In Section 3.1, "Generating FL Clients," it is unclear how the proposed method differs from stratified sampling in addressing imbalanced classes. This distinction needs to be clarified.  

5/A general architecture diagram for the proposed solution must follow the algorithms and explanation text. This would provide a clearer understanding of the overall approach.  

6/Typos such as "Numbero f Clusters" need to be corrected. Please carefully proofread the manuscript.  

7/The detection of attack records should ideally be independent of adjacent records. Furthermore, as the authors mention, data is distributed randomly to clients. In this context, the use of a long-term dependency model like BiLSTM for attack detection appears questionable. Please justify its suitability for this task.  

 8/It is unclear if all features were included in the training and testing process. Specifically, how were features like identifiers (e.g., ID, ime-to-Live (TTL)-based features) handled? Please provide a detailed explanation.  

9/A comparison of the proposed solution's results with those of related works is missing. Including this comparison is critical to assess the contributions and performance improvements of the proposed approach.  

Author Response

Thank you very much for taking the time to review this manuscript, we really appreciate your valuable comments. Please find detailed responses in the attached file.

Round 2

Reviewer 2 Report

Comments and Suggestions for Authors

The authors have addressed most of my comments, and I have no further remarks.